# Circulating Angiogenic Factors and Ischemic Diabetic Foot Syndrome Advancement—A Pilot Study

**DOI:** 10.3390/biomedicines11061559

**Published:** 2023-05-27

**Authors:** Martyna Schönborn, Iwona Gregorczyk-Maga, Krzysztof Batko, Katarzyna Bogucka, Mikołaj Maga, Anna Płotek, Patrycja Pasieka, Krystyna Słowińska-Solnica, Paweł Maga

**Affiliations:** 1Department of Angiology, Faculty of Medicine, Jagiellonian University Medical College, 30-663 Krakow, Poland; maga.pawel@gmail.com; 2Doctoral School of Medical and Health Sciences, Jagiellonian University, 31-007 Krakow, Poland; 3Clinical Department of Angiology, University Hospital in Krakow, 30-688 Krakow, Poland; kat.bogucka1@gmail.com (K.B.); mikolaj.maga@gmail.com (M.M.); anplotek@gmail.com (A.P.); 4Faculty of Medicine, Institute of Dentistry, Jagiellonian University Medical College, 30-663 Krakow, Poland; iwona.gregorczyk-maga@uj.edu.pl; 5Department of Research and Development, Medicine Economy Law Society (MELS) Foundation, 30-040 Krakow, Poland; batko.krzysztof@gmail.com; 6Department of Rehabilitation in Internal Medicine, Faculty of Health Sciences, Jagiellonian University Medical College, 31-008 Krakow, Poland; 7Department of Dermatology, Faculty of Medicine, Jagiellonian University Medical College, 31-008 Krakow, Poland; patrycja.laczak25@gmail.com; 8Department of Clinical Biochemistry, Jagiellonian University Medical College, 30-663 Krakow, Poland; krystyna.slowinska-solnica@uj.edu.pl

**Keywords:** angiogenic factors, angiogenesis, diabetic foot syndrome, microcirculation, peripheral arterial disease

## Abstract

Despite clear evidence of inadequate angiogenesis in ischemic diabetic foot syndrome (DFS) pathogenesis, angiogenic factor level changes in patients with ischemic DFS remain inconsistent. This study aimed to assess circulating angiogenic factors concerning ischemic DFS advancement and describe their relationships with patients’ clinical characteristics, microvascular parameters, and diabetic control. The study included 41 patients with ischemic DFS (67.3 (8.84) years; 82.9% males). Angiogenic processes were assessed by identifying circulating concentrations of five pro- and two anti-angiogenic factors. We found that penetrating ulcers were related to a significantly higher FGF-2 level (8.86 (5.29) vs. 5.23 (4.17) pg/mL, *p* = 0.02). Moreover, plasma FGF-2 showed a significant correlation with the SINBAD score (r = 0.32, *p* = 0.04), platelet count (r = 0.43, *p* < 0.01), white cell count (r = 0.42, *p* < 0.01), and age (r = −0.35, *p* = 0.03). We did not observe any significant linear relationship between the studied biomarkers and microcirculatory parameters, nor for glycemic control. In a univariate analysis using logistic regression, an increase in plasma FGF-2 was tied to greater odds of high-grade ulcers (OR 1.16; 95% CI 1.02–1.38, *p* = 0.043). This suggests that circulating FGF-2 may serve as a potential biomarker for predicting DFU advancement and progression. It is necessary to conduct further studies with follow-up observations to confirm this hypothesis.

## 1. Introduction

Diabetic foot syndrome (DFS) is one of the most common, chronic, and complex complications of diabetes mellitus (DM). It is estimated that the pooled worldwide prevalence of diabetic foot ulcerations (DFUs) is 6% [1] and the lifetime risk of DFUs in diabetes patients even reaches 34% with an annual incidence of 2% [2]. Based on the International Working Group of the Diabetic Foot (IWGDF), a diabetic foot may be characterized by infection, ulceration, or destruction of foot tissue in a person with diagnosed diabetes [3]. The pathophysiology of DFS is complex, but the underlying mechanism can be presented as a triad of neuropathy, trauma with secondary infection, and arterial insufficiency from peripheral arterial occlusive disease (PAD) [4]. Peripheral neuropathy results in inherent muscle atrophy, which leads to alterations in the function and anatomy of the affected foot muscles. The combination of repetitive walking trauma, reduced sensation, and impaired proprioception increases the susceptibility to skin damage, ultimately leading to the development of ulcers and subsequent infections. Alterations in glucose metabolism contribute to endothelial damage; hyperlipidemia; enhanced platelet viscosity and activity; and, over time, the progression of atherosclerotic plaques [4].

DFS with concomitant PAD is classified as ischemic diabetic foot syndrome. It is especially connected to an unpredictable and often poor prognosis [5]. In diabetic patients, PAD is frequently asymptomatic and occurs early with rapid progression because of multiple metabolic aberrations in DM. It is also more diffuse and affects distal limb arteries, such as tibial and peroneal arteries, compared with non-diabetic PAD [4,6]. Insufficient skin perfusion in patients with DFUs is independently associated with non-healing and risk of amputation [7,8]. Based on epidemiological studies, approximately 50–70% of all lower limb amputations worldwide are related to diabetes, and according to the estimates by the IWGDF, they are performed every 30 s [9].

In tissue healing, a substantial balance between cell migration, proliferation, and remodeling and precise responses to inflammatory mediators and angiogenic and growth factors is necessary [9]. One of the cornerstones of proper wound granulation and closure is adequate vascular flow and its sufficiency and proliferation during the angiogenesis process. New vessel creation is regulated by a delicate balance between pro-angiogenic and anti-angiogenic factors. The critical activator of the whole process is hypoxia resulting from blood vessel damage [10]. Endothelial cells, essential for the formation of new blood vessels, secrete paracrine molecules known as angiocrine factors. These factors include fibroblast growth factor (FGF), insulin-like growth factor (IGF), and platelet-derived growth factor (PDGF) [11]. During the first phase of angiogenesis, pro-angiogenic factors such as fibroblast growth factor 2 (FGF-2) and vascular endothelial growth factor A (VEGF-A) are expressed locally to stimulate capillaries to form immature loops and branches. Then, a switch to anti-angiogenic factors occurs, leading to vessel maturation and regression [9,10,11,12]. In DFS, each phase of the healing process may be unco-ordinated, incomplete, or postponed [9], and insufficient angiogenesis plays a significant role in the pathogenesis of non-healing wounds. The underlying cause of numerous microvascular and macrovascular complications in diabetic patients can be persistent hyperglycemia. These complications can ultimately impact the process of angiogenesis. Endothelial cells exposed to hyperglycemia may result in impaired function, compromising their integrity and rendering them more vulnerable to apoptosis, detachment, and subsequent circulation within the bloodstream. In diabetic wounds, insufficient angiogenesis leads to reduced vascularity and capillary density, resulting in significant delays in wound closure compared with non-diabetic wounds [10]. Disruptions in angiogenesis also have a significant impact on the homeostasis of the skeletal system, which can also be disturbed among individuals with DFS. Endothelial cells secrete a variety of substances with paracrine and autocrine activity and regulate bone remodelling via the cell signalling networks of ligand–receptor complexes [13].

Other integral components of pathologic processes in DFUs are structural and functional changes in the microvasculature [14]. The most notable morphologic alterations involve capillary basement membrane thickening, reductions in capillary size, and pericyte degeneration initiated by increased hydrostatic pressure and shear forces [15]. These processes are accompanied by an impaired ability to vasodilate in response to stress or injury and a maldistribution of blood flow between subpapillary arteriovenous shunts and nutritional capillaries [14]. Nevertheless, available studies have not drawn firm conclusions about the degree of microvascular function improvement required to improve DFU healing [16].

Despite compelling evidence suggesting inadequate angiogenesis and microvascular dysfunction in the pathogenesis of diabetic foot syndrome, the alterations in angiogenic factor levels in the bloodstream or wound materials among individuals with DFUs remain inconsistent [17]. Because of the clinical heterogeneity of ulcerations, accurately monitoring the healing process and predicting the probability of limb amputation poses a challenge. To date, there are no reliable risk prediction models. Therefore, developing such a model or deriving biochemical predictors of poor prognosis is highly interesting.

This study aimed to assess circulating pro- and anti-angiogenic factors concerning DFS advancement and describe potential relationships between clinical characteristics, microcirculatory parameters, and diabetic control among patients with ischemic DFS.

## 2. Materials and Methods

### 2.1. Study Design and Population

A single-centered, cross-sectional study involving patients recruited from the Clinical Department of Angiology between February 2021 and May 2022 was conducted. The inclusion criteria were as follows: (1) subjects aged 40–80 years old with DM type 2, (2) ischemic DFS with active lower limb ulcerations, and (3) concomitant critical limb ischemia due to PAD (category 5 or 6 in the Rutherford classification). Patients with myocardial infarction or stroke within the last 6 months, diagnosed Charcot’s foot, acute lower limb ischemia within the previous 3 months, chronic kidney disease with eGFR < 45 mL/min/1.73 m^2^, or neoplasm diagnosed within 5 years were excluded from the study. The study did not include subjects with chronic infectious diseases (e.g., hepatitis C virus, or human immunodeficiency virus) or autoimmune comorbidities (e.g., rheumatoid arthritis, or scleroderma).

### 2.2. Data Collection

Information on chronic disorders, smoking status, and medications was obtained during medical history collection. All data and results were stored using a certificated tool for data collection (Medrio EDC). Hypertension was defined as an SBP ≥ 140 mm Hg and/or DBP ≥ 90 mm Hg, or receiving antihypertensive treatment. Body mass index (BMI) was calculated by dividing weight in kilograms and height in meters squared. Hemoglobin level, platelet count (PLT), white blood cell count (WBC), creatinine, C-reactive protein (CRP), glycated hemoglobin (HbA1c), total cholesterol (TC), triglyceride (TG), high-density lipoprotein cholesterol (HDL-C), and low-density lipoprotein cholesterol (LDL-C) concentrations were measured on the day of admission to the hospital and were extracted from the hospital laboratory system. The extra blood samples were collected from all patients and tested for the biomarker levels listed below. All samples were collected after a 12 h fasting period.

### 2.3. Angiogenic Factor Assessment

Angiogenic processes were assessed by identifying concentrations of 7 physiologically produced circulating biomarkers with well-documented pro- or anti-angiogenic characteristics. The five pro-angiogenic biomarkers were vascular endothelial growth factor A (VEGF-A), soluble form of vascular endothelial growth factor receptor 2 (VEGF-R2, also called sVEGF-R2), fibroblast growth factor 2 (FGF-2) (also called basic fibroblast growth factor (bFGF)), placental growth factor (PlGF), and platelet-derived growth factor-BB (PDGF-BB). The two assessed anti-angiogenic factors comprised pigment epithelium-derived factor (PEDF) and angiopoietin-1 (Ang-1). The main material for angiogenic factor assessment was plasma separated from ethylenediaminetetraacetic (EDTA) whole-fasting blood samples after centrifugation at 3000 rpm for 15 min. All samples were stored at −80 °C until analysis. Human VEGF-A and VEGF-R2 concentrations (pg/mL) were determined using an enzyme-linked immunosorbent assay with commercially available ELISA kits (Thermo Fisher Scientific, Inc., Waltham, MA, USA; Cat. No. BMS277 and BMS2019). Human PDGF-BB, ANG-1, FGF, and PIGF levels (pg/mL) were measured via the quantitative sandwich enzyme immunoassay technique (R&D Systems, Minneapolis, MN, USA; Cat. No. DBB00, DANG10, DFB50, and DPG00). Human PEDF concentration (µg/mL, but data shown in ng/mL) was determined via an immunoenzymatic method using a commercially available ELISA kit (BioVendor—Laboratorni Medicina a.s., Brno, Czech Republic, Cat. No. RD191114200R). 

### 2.4. Assessment of DFS Advancement

The category of chronic limb ischemia was evaluated using the symptomatic Rutherford classification, which defines category 5 as minor tissue loss and category 6 as major tissue loss extending above the transmetatarsal level [18]. The SINBAD classification system and the WIfI classification developed by the Society of Vascular Surgery were used to assess diabetic foot syndrome advancement and amputation risk estimation. WIfI comprises an evaluation of the wound presence, insensitivity of ischemia, and foot infection with 4 grades of severity for each category. It can also determine 64 possible clinical combinations with an estimated risk of amputation at 1 year (very low, low, moderate, or high) [19]. In the SINBAD classification, six elements are graded: ulcer site (forefoot vs. midfoot/hindfoot), ischemia (at least one pulse palpable vs. evidence of ischemia), neuropathy (absent vs. present), bacterial infection (absent vs. present), area (ulcer < 1 cm^2^ vs. ≥ 1 cm^2^), and depth (confined to the skin and subcutaneous tissue vs. reaching muscle, tendons, or more profound). The components of the SINBAD classification can be summed to produce a score between 0 and 6. It divides ulcers into 3 groups: low grade (0–2), moderate grade (3–4), and high grade (5–6) [20].

### 2.5. Hemodynamic Parameters of Lower Limb Arteries

The status of lower limb ischemia was assessed using an estimated ankle–brachial index (ABI) and toe–brachial index (TBI). Both examinations included systolic blood pressure measurements of the brachial arteries, ankle arteries, and toe capillaries. These were performed using a sphygmomanometer, an 8 MHz blind Doppler flow detector, and digital plethysmography. ABI was determined by dividing the higher systolic blood pressure measured at the dorsalis pedis or posterior tibial artery and the higher systolic blood pressure measured at the right or left brachial artery. In addition to ABI, systolic blood pressure on the toe was measured with a plethysmographic sensor and TBI was estimated. TBI was assessed by dividing the systolic blood pressure measured at the toe capillaries and the higher systolic blood pressure at the right or left brachial artery. All examinations were performed in a controlled environment with a room temperature between 21 and 23 degrees Celsius. The measurements were taken after a 15 min rest in a supine position with the limbs parallel to the body.

### 2.6. Microcirculation Assessment

Microcirculation was evaluated using laser Doppler flowmetry (LDF) and the transcutaneous oximetry test (tcpO2). Both measurements were made with the Periflux 6000 (Perimed AB, Järfälla, Sweden), equipped with thermostatic laser Doppler probes to precisely heat the tissue at the measurement site and a modified Clark’s polarographic oxygen sensor. The probes were stabilized on the dorsal aspect of the distal part of the foot, excluding skin with necrosis or inflammation, bone prominence, or a superficial tendon. Baseline microvascular blood flow was measured for 5 min and expressed in arbitrary perfusion units (PU). The tcpO2 examination was conducted for 20 min or until the curve on the graph flattened and was presented in mmHg. The results reflect microvascular flow, including capillaries, arterioles, venules, and shunts, as well as microcirculation blood perfusion, metabolic activity, oxyhemoglobin dissociation, and tissue oxygen partial pressure. Both tests were conducted in a room at a temperature of 21–23 degrees Celsius. Testing was preceded by a 15 min resting period, during which the patient lay in a comfortable position. 

### 2.7. Statistical Analysis and Sample Size Calculation

Statistical analysis was performed in R 4.2.2 (R Team, R Statistical Foundation, Vienna, Austria) using publicly available packages (tidyverse, arsenal, rstatix, and ggpubr). Continuous variables were summarized as mean and standard deviation, with categorical variables as counts and proportions. Right-skewed variables were log-transformed before analyses. Distribution was assessed using qqplots. Continuous and nominal variables were compared using a *t*-test and Fisher’s test, respectively. Pearson’s correlation coefficient was used to assess linear relationships. Tests were two-tailed and a *p*-value < 0.05 was treated as significant.

We assumed a moderate (r = 0.2 to r = 0.4) association between circulating angiogenesis regulating factors and clinical characteristics. Given a significance level of 0.05 and a power set at 0.80, the required sample size for a one-sided hypothesis is at least 37 patients.

### 2.8. Ethical Aspects

This research is not in conflict with any ethical norms and regulations in research studies on humans. All patients signed informed consent forms to participate in the study. The regulations of the Declaration of Helsinki were used to prepare the protocol for this study, and the proper consent for this study was obtained from the constituted committee for human subjects or animal research at Jagiellonian University Medical College (decision number 1072.6120.129.2020).

## 3. Results

### 3.1. Demographic and Clinical Characteristics of Subjects

In this sample (n = 41), the mean (SD) age of the patients was 67.3 (8.84) years, with an average BMI of 27.55 (3.47) and substantial male sex predominance (n = 34, 82.9%). The cardiovascular disease risk of the population was very high: 29 were smokers (70.7%), 37 had hypertension (90.2%), 15 had coronary artery disease (36.6%), 10 had congestive heart failure (24.4%), eight had atrial fibrillation (19.5%), five had a prior history of stroke (12.2%), and four had chronic kidney disease (9.8%). All patients presented with Rutherford categories 5 (31, 75.6%) or 6 (10, 24.4%). Almost half of the patients (19, 46.3%) had a prior leg amputation, distal (16, 84.2%) or proximal (3, 15.8%) to the ankle. According to the SINBAD classification, wound infection was recorded for 23 patients (56.1%), 21 had a penetrating ulcer (51.2%), and 15 patients had a forefoot wound location (36.6%). The mean SINBAD score was 4.29 (1.21). Regarding the WIfI classification system, 30 patients (93.8%) were in the highest ischemic category (tcpO2 < 30 mm Hg). Most patients were in the high-risk amputation group (27, 65.9%), while 11 patients (26.8%) were categorized as moderate risk.

### 3.2. Comparison of Subjects’ Characteristics According to Ulcer Features

The comprehensive clinical characteristics of subjects divided into two groups based on ulceration depth (limited vs. penetrating) are presented in Appendix A. A comparison of blood count and biochemical tests between the groups showed that patients with penetrating ulcers had a significantly lower hemoglobin level (13.79 (2.10) vs. 12.28 (1.66), *p* = 0.01).

Regarding parameters reflecting systemic inflammation and circulating angiogenic factor levels, we observed that penetrating ulcers were related to significantly higher FGF-2 (8.86 (5.29) vs. 5.23 (4.17) pg/mL, *p* = 0.02), PLT (5.81 (0.28) vs. 5.50 (0.28), *p* < 0.01), and CRP levels (2.73 (1.46) vs. 1.35 (1.08), *p* < 0.01) (see Table 1).

Based on the analysis of hemodynamic parameters and microvascular status, we observed no correlations between ulceration depth and ABI, TBI, or tcpO2. However, the results revealed that individuals with penetrating ulcers exhibited significantly elevated levels of LDF (14.48 (5.07) vs. 11.57 (3.56) PU, *p* = 0.04) (see Table 2).

### 3.3. Comparison of Angiogenic Factors and Clinical Characteristics According to Patients’ Age

Generally, age did not significantly impact most of the clinical features (Appendix A). However, in patients aged 40–68, the levels of FGF-2 were significantly higher compared with those in patients over 68 years old (8.83 (5.78) vs. 5.26 (3.43), *p* = 0.02). In terms of glycemic control, HbA1c levels were significantly higher in the younger group compared with patients over 68 years old (8.47 (1.72) vs. 7.30 (1.27), *p* = 0.02).

Figure 1, Figure 2, Figure 3, Figure 4, Figure 5, Figure 6, Figure 7, Figure 8, Figure 9, Figure 10, Figure 11, Figure 12, Figure 13, Figure 14 and Figure 15 present the relationships between clinical characteristics, glycemic control, and all angiogenic factors for the entire group of patients.

### 3.4. Relationships between Pro- and Anti-Angiogenic Factors, Microvascular Status, and Glycemic Control

We did not observe any significant linear relationship between the studied angiogenic biomarkers and microcirculatory parameters (LDF, tcpO2) (Figure 2, Figure 4, Figure 6, Figure 8, Figure 10, Figure 12, and Figure 14). There was also no impact from glycemic control (defined as HbA1c ≤ 8% or HbA1c > 8%) on the concentrations of the angiogenic markers, as shown in Figure 15.

### 3.5. Relationships between Plasma FGF-2 and Clinical Characteristics

We found a significant correlation between circulating FGF-2 and several clinical characteristics (Figure 1 and Figure 2). It was associated with penetrating ulcer presence (mean (SD); 8.86 (5.29) vs. 5.23 (4.17), *p* = 0.02) and smoker status (mean (SD); 8.04 (5.27) vs. 4.80 (3.77), *p* = 0.04) but not sex (*p* = 0.54), hypertension (*p* = 0.55), coronary artery disease (*p* = 0.93), prior MI (*p* = 0.87), or CKD (*p* = 0.81). Plasma FGF-2 showed a significant correlation with platelet count (r = 0.43, *p* < 0.01), white cell count (r = 0.42, *p* < 0.01), age (r = −0.35, *p* = 0.03), and the SINBAD score (r = 0.32, *p* = 0.04). No significant linear association was observed for BMI (*p* = 0.85), hemoglobin (*p* = 0.18), eGFR (*p* = 0.94), HbA1c (*p* = 0.48), TC (*p* = 0.52), LDL (*p* = 0.29), HDL (*p* = 0.80), TG (*p* = 0.53), CRP (*p* = 0.24), tcpO2 (*p* = 0.26), LDF (*p* = 0.76), ABI (*p* = 0.06), or the WIFI score (*p* = 0.13). Using different antidiabetic drugs was not associated with circulating FGF-2 concentrations.

In the univariate analysis using logistic regression, an increase in plasma FGF-2 was tied to greater odds of high-grade ulcers according to the SINBAD classification (OR 1.16; 95% CI 1.02–1.38, *p* = 0.043). Using a multivariable model with log-transformed CRP and FGF-2 concentrations, we observed an improvement in model fit. However, only the relationship with CRP remained significant (OR 1.98; 95% CI 1.15–3.91, *p* = 0.025). We further assessed the predictive performance of this simple model using three thresholds (0.3, 0.5, and 0.7), which yielded a corresponding sensitivity and specificity of 94.74%/27.78%, 68.42%/72.22%, and 42.11%/88.89%, respectively.

## 4. Discussion

The key finding of the present study is that FGF-2 was significantly associated with ulceration depth and showed a significant correlation with the SINBAD score. We found that the mean FGF-2 level was higher among subjects with penetrating ulcers and a higher SINBAD score. Moreover, an increase in plasma FGF-2 was tied to greater odds of high-grade ulcers. In this study, among the analyzed angiogenic factors, only plasma FGF-2 concentrations could differentiate patients with penetrating ulceration. These results may suggest a hypothesis that elevated systemic FGF-2 could promote more advanced DFU. To the best of our knowledge, currently, there is a lack of research regarding the significance of FGF-2 as a predictor in the advancement and progression of DFS.

Our results are consistent with existing data on circulating FGF-2 concentration differences between DFU patients and healthy subjects. Kulwas et al. showed increased circulating FGF-2 levels in DFU compared with individuals without DFU [21]. Moreover, in Tecilazih’s study, median values of serum FGF-2 were the highest for DFU patients, but no statistical analysis was conducted, and, thus, no definite conclusions can be drawn [22]. Gui et al. obtained comparable findings, as they noted an increase in the concentration of circulating FGF-2 among diabetic individuals in contrast with the control group [23]. However, these findings are in contrast with the general knowledge about FGF-2 angiogenic and mitogenic effects, as well as its role in dermal fibroblast migration and stimulating the proliferation phase in wound healing [24]. FGF-2 is one of the most widely studied factors in patients with DFS, and its use as a topically administered drug in non-healing ulcers has been the main topic in recent trials [24,25].

On the other hand, circulating FGF-2 plays a role in inflammatory processes and atherosclerotic lesion growth. It may stimulate intimal thickening, intraplaque angiogenesis, and the proliferation of vascular smooth muscle cells [26,27]. Furthermore, local hyperglycemia and advanced glycation end products (AGEs) in diabetic patients may promote glycosylated FGF-2 production, which inhibits the proliferation of endothelial cells and has negative effects on wound healing [21,28]. The glycation of angiogenic factors is associated with unresponsiveness to them [29]. One study showed that increased plasma FGF-2 levels might even be a risk biomarker for coronary heart disease occurrence in adult men with DM type 2 [30]. Therefore, it is crucial to differentiate between its various effects depending on the analyzed material. While its local deficiency in healing tissue is associated with insufficient angiogenesis and delayed healing, its elevated circulating form may have unfavorable effects on the course of DFS and cardiovascular disease progression in patients with DM. The potential double pathway of FGF-2 action on impaired wound healing is presented in Figure 16.

We observed that FGF-2 concentrations were more likely to be higher among patients who were ever smokers (active and past smokers). These results are consistent with observations obtained from bovine models. In aortic endothelial and smooth muscle cells, bFGF release after nicotine stimulation was greater than in controls [31,32]. Similar results were obtained from previous studies using human osteoblasts and smooth muscle cells [33,34]. We showed that plasma FGF-2 significantly correlated with PLT and WBC. These findings align with a study on platelet activation in burn patients, where the amount of FGF-2 per platelet was constant for 21 days [35]. The FGF family has a known influence on inflammatory processes [36]. A few studies have shown that FGF-2 is upregulated in inflammatory disorders and may act as an immunoregulator of T-lymphocytes, neutrophils, macrophages, and monocytes [36,37,38]. These findings may support our result regarding the increase in FGF-2 levels with WBC.

Additionally, we presented a reverse relationship between FGF-2 and age. This phenomenon remains a subject of discussion according to Harely M. et al., who found a progressive decrease in FGF-2 with age in human mesenchyme-derived progenitor cell cultures [39]. However, there are also studies in which relationships between FGF-2 levels and age were not present [21,40].

Although there is convincing evidence indicating insufficient angiogenesis in the development of DFS, the changes in circulating angiogenic factor levels in the blood or wound materials among patients with DFUs are inconclusive. Several studies have emphasized the significance of reduced levels of pro-angiogenic factors such as VEGF-A and FGF-2 in wound material, as well as the decreased expression of PlGF in the development of DFUs [17,41]. Additionally, research has validated that increased levels of anti-angiogenic factors, such as PEDF, play a significant role in impeding angiogenesis and the healing of wounds [42]. Numerous inconsistencies in the field remain, such as the paradox of impaired wound healing despite the presence of high levels of pro-angiogenic factors or fluctuations in the expression of pro-angiogenic factors during the healing process.

According to ulceration features, our study found that ulceration depth had significant relationships with certain clinical characteristics. Penetrating ulcers were tied to lower hemoglobin levels, higher platelet counts, and CRP concentrations, which are reflective of an inflammatory state with chronic iron restriction. Our observation is consistent with conclusions obtained from the other studies. A recent systematic review showed that lower hemoglobin levels are associated with DFU advancement, non-healing ulcers, amputation, and mortality [43]. Further, Wang et al. showed that CRP is an essential biomarker in differentiating grade 1 from grade 2 DFUs [44]. CRP has been proposed as a prognostic marker for DFU healing [22].

Only a few studies have described the effect of HbA1c levels on ulcer healing [45,46]. Chen et al. showed that HbA1C might be one of the risk factors for early DFU development in diabetic patients [47]. Nevertheless, we did not observe any significant correlations between the studied biomarker panel and glycemic control, nor microcirculatory parameters. While a potential linear trend between vascular endothelial growth factor receptor 2 (VEGF-R2) and both tcpO2 and LDF measurements was observed, whether such a relationship exists requires confirmation in larger samples.

The main strength of this study is a relatively large number of assessed circulating angiogenic factors, which were evaluated in a homogeneous group of patients with ischemic DFS. Despite concerns regarding study power, we observed that FGF-2 might be tied to specific clinical characteristics reflecting DFS advancement. Nevertheless, our study does have several limitations. The main one includes the small sample size, the monocentric character of the study, and the lack of a healthy control group. This cross-sectional study examined forty-one ischemic DFS patients who represent a homogenous sample because of highly restrictive recruitment criteria (in contrast to earlier studies [20,21]). Nevertheless, as a pilot study, it enabled the initial identification of the most promising angiogenic factor in terms of its predictive potential.

## 5. Conclusions

This study indicated a significant correlation between elevated plasma FGF-2 levels and ulceration depth, as well as with the SINBAD score, in ischemic DFS patients. Penetrating ulcers were related to significantly higher plasma FGF-2 concentrations, and an increase in FGF-2 was tied to greater odds of high-grade ulcerations. This suggests that FGF-2 may serve as a potential biomarker for predicting DFU advancement and progression. Future research with follow-up observations should investigate changes in circulating FGF-2 over time and their relationships with DFU healing and lower limb amputation rates to verify its predictive value.

## Figures and Tables

**Figure 1 biomedicines-11-01559-f001:**
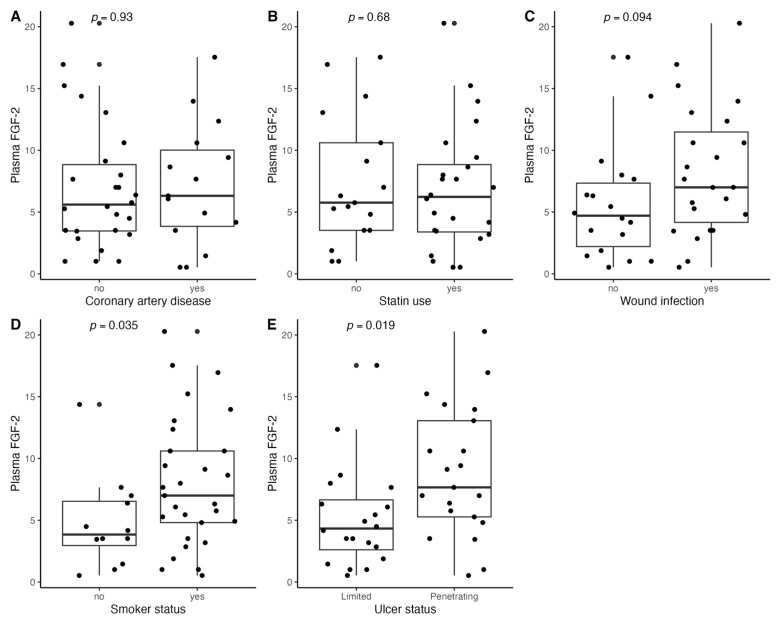
Boxplots with jittered data illustrating the relationships between plasma FGF-2 and (**A**) coronary artery disease, (**B**) statin use, (**C**) wound infection, (**D**) smoking status (yes—active or past smoker), and (**E**) ulcer depth FGF-2—fibroblast growth factor 2.

**Figure 2 biomedicines-11-01559-f002:**
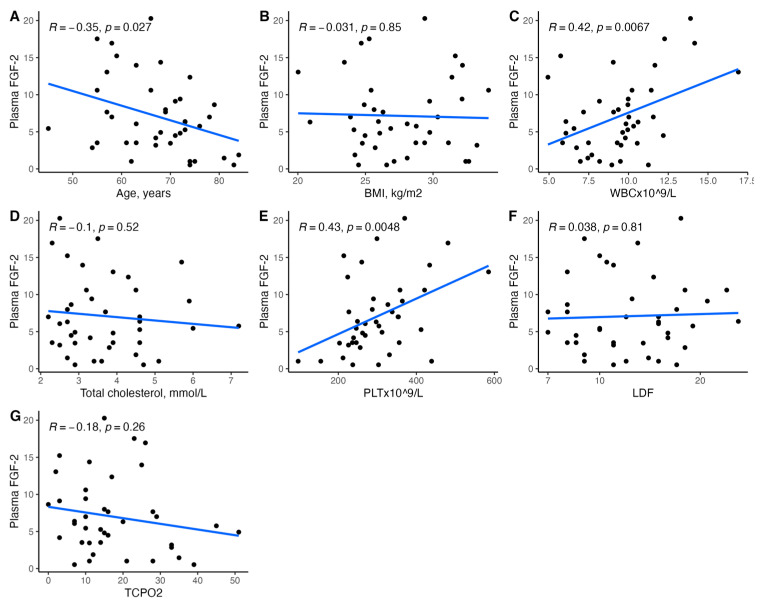
Scatterplots with a linear fit illustrating the relationships between plasma FGF-2 and (**A**) age, (**B**) BMI, (**C**) WBC, (**D**) total cholesterol, (**E**) PLT, (**F**) LDF, and (**G**) tcpO2. FGF-2—fibroblast growth factor 2, BMI—body mass index, WBC—white blood cells count, PLT—platelet count, LDF—laser Doppler flowmetry, and tcpO2—transcutaneous oximetry test.

**Figure 3 biomedicines-11-01559-f003:**
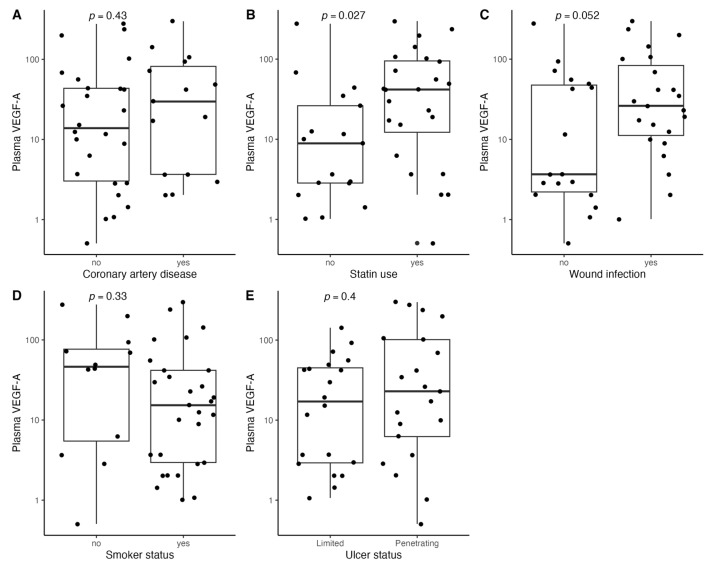
Boxplots with jittered data illustrating the relationships between plasma VEGF-A and (**A**) coronary artery disease, (**B**) statin use, (**C**) wound infection, (**D**) smoking status (yes—active or past smoker), and (**E**) ulcer depth VEGF-A—vascular endothelial growth factor A.

**Figure 4 biomedicines-11-01559-f004:**
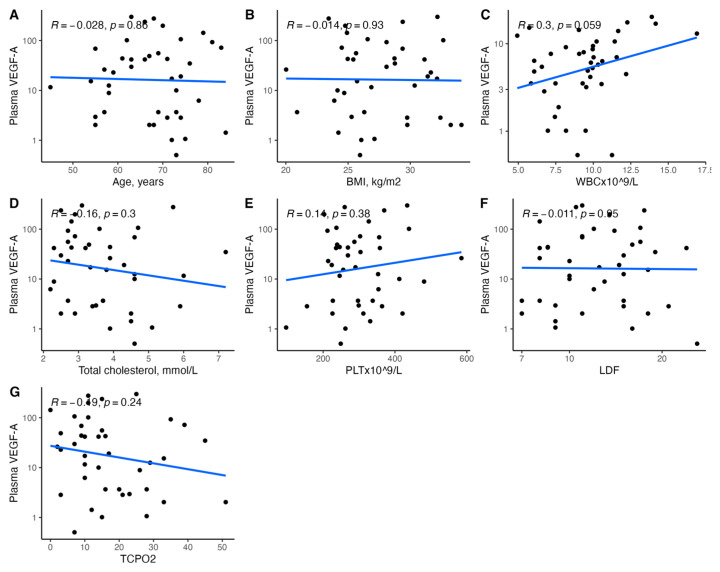
Scatterplots with a linear fit illustrating the relationships between plasma VEGF-A and (**A**) age, (**B**) BMI, (**C**) WBC, (**D**) total cholesterol, (**E**) PLT, (**F**) LDF, and (**G**) tcpO2. VEGF-A—vascular endothelial growth factor A, BMI—body mass index, WBC—white blood cells count, PLT—platelet count, LDF—laser Doppler flowmetry, and tcpO2—transcutaneous oximetry test.

**Figure 5 biomedicines-11-01559-f005:**
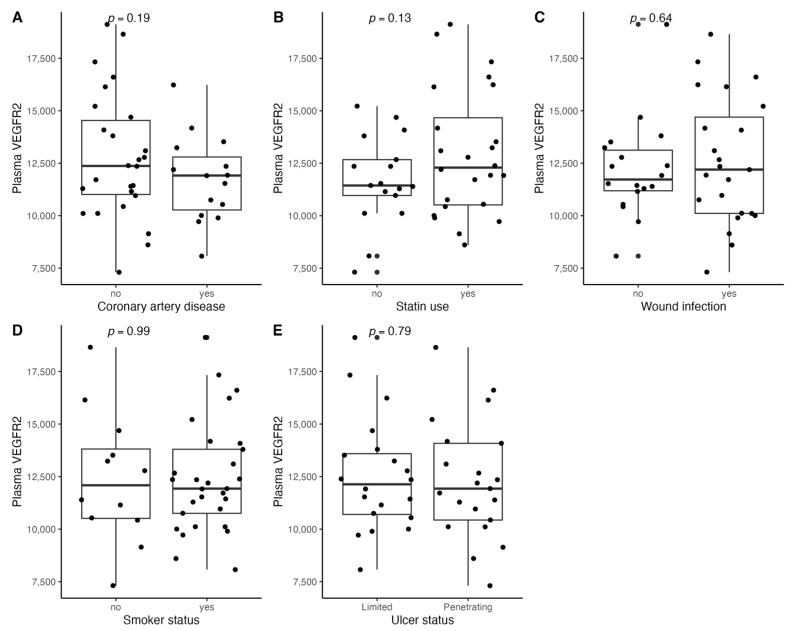
Boxplots with jittered data illustrating the relationships between plasma VEGF-R2 and (**A**) coronary artery disease, (**B**) statin use, (**C**) wound infection, (**D**) smoking status (yes—active or past smoker), and (**E**) ulcer depth VEGF-R2—vascular endothelial growth factor receptor 2.

**Figure 6 biomedicines-11-01559-f006:**
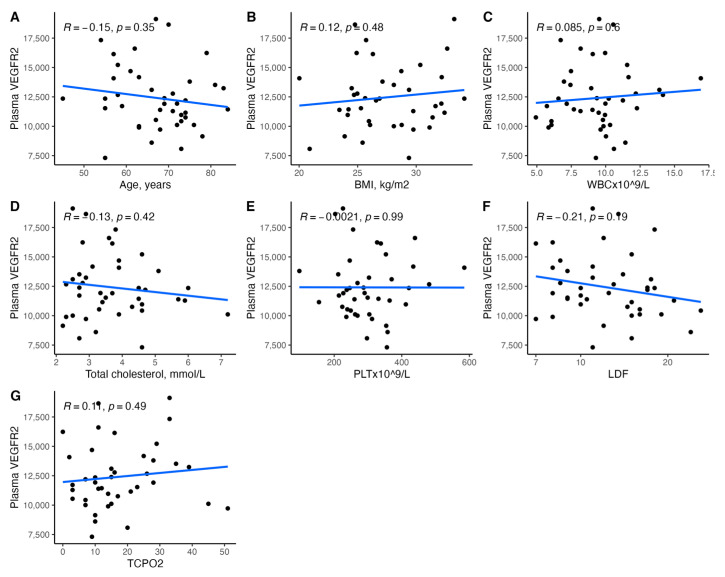
Scatterplots with a linear fit illustrating the relationships between plasma VEGF-R2 and (**A**) age, (**B**) BMI, (**C**) WBC, (**D**) total cholesterol, (**E**) PLT, (**F**) LDF, and (**G**) tcpO2. VEGF-R2—vascular endothelial growth factor receptor 2, BMI—body mass index, WBC—white blood cells count, PLT—platelet count, LDF—laser Doppler flowmetry, and tcpO2—transcutaneous oximetry test.

**Figure 7 biomedicines-11-01559-f007:**
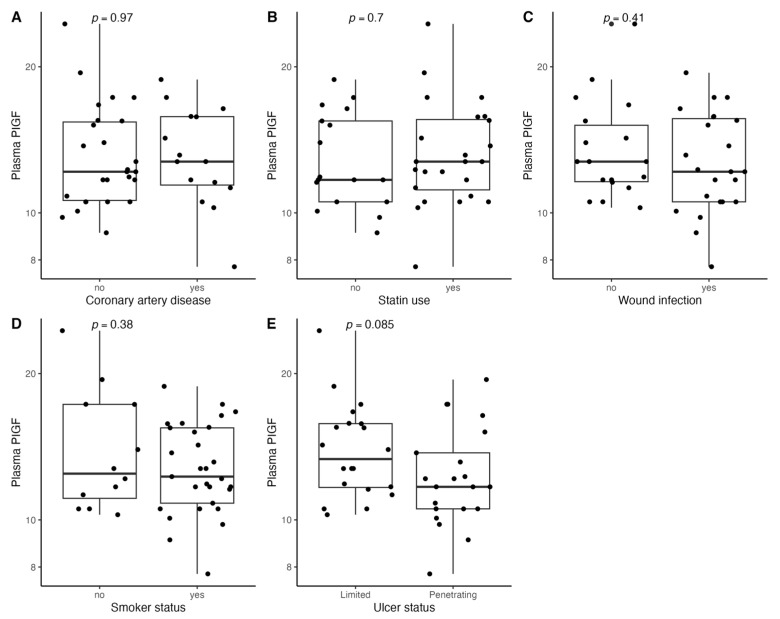
Boxplots with jittered data illustrating the relationships between plasma PlGF and (**A**) coronary artery disease, (**B**) statin use, (**C**) wound infection, (**D**) smoking status (yes—active or past smoker), and (**E**) ulcer depth PlGF—placental growth factor.

**Figure 8 biomedicines-11-01559-f008:**
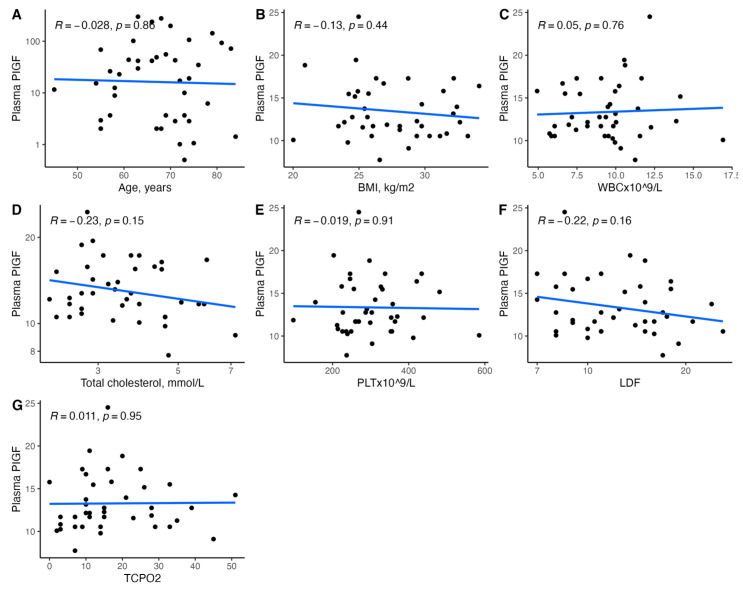
Scatterplots with a linear fit illustrating the relationships between plasma PlGF and (**A**) age, (**B**) BMI, (**C**) WBC, (**D**) total cholesterol, (**E**) PLT, (**F**) LDF, and (**G**) tcpO2. PlGF—placental growth factor, BMI—body mass index, WBC—white blood cells count, PLT—platelet count, LDF—laser Doppler flowmetry, and tcpO2—transcutaneous oximetry test.

**Figure 9 biomedicines-11-01559-f009:**
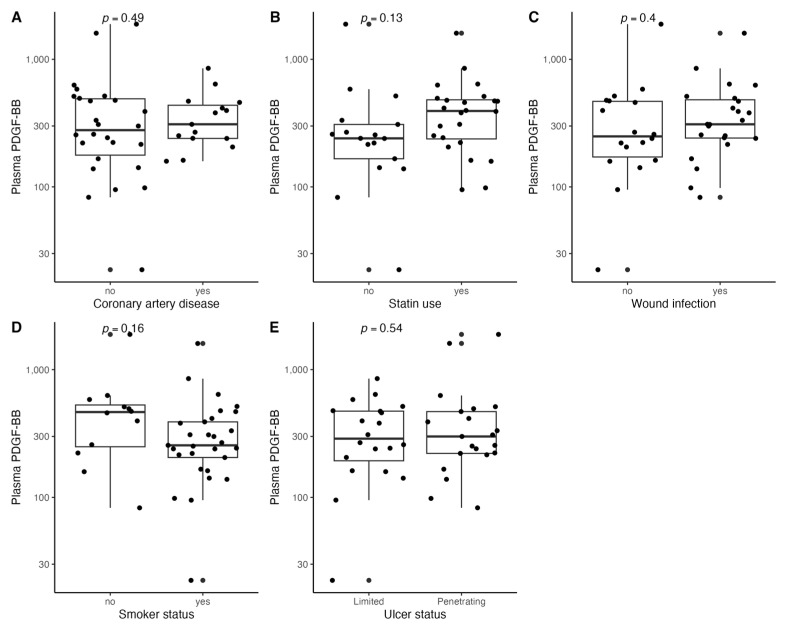
Boxplots with jittered data illustrating the relationships between plasma PDGF-BB and (**A**) coronary artery disease, (**B**) statin use, (**C**) wound infection, (**D**) smoking status (yes—active or past smoker), and (**E**) ulcer depth PDGF-BB—platelet-derived growth factor-BB.

**Figure 10 biomedicines-11-01559-f010:**
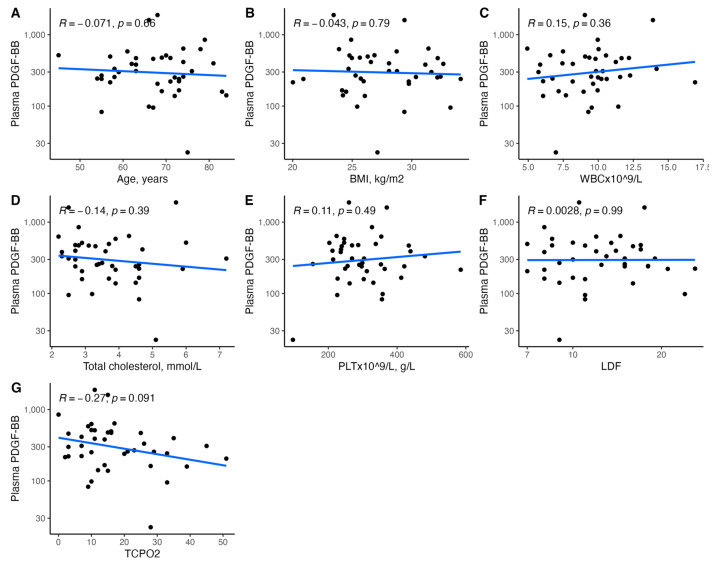
Scatterplots with a linear fit illustrating the relationships between plasma PDGF-BB and (**A**) age, (**B**) BMI, (**C**) WBC, (**D**) total cholesterol, (**E**) PLT, (**F**) LDF, and (**G**) tcpO2. PDGF-BB—platelet-derived growth factor-BB, BMI—body mass index, WBC—white blood cells count, PLT—platelet count, LDF—laser Doppler flowmetry, and tcpO2—transcutaneous oximetry test.

**Figure 11 biomedicines-11-01559-f011:**
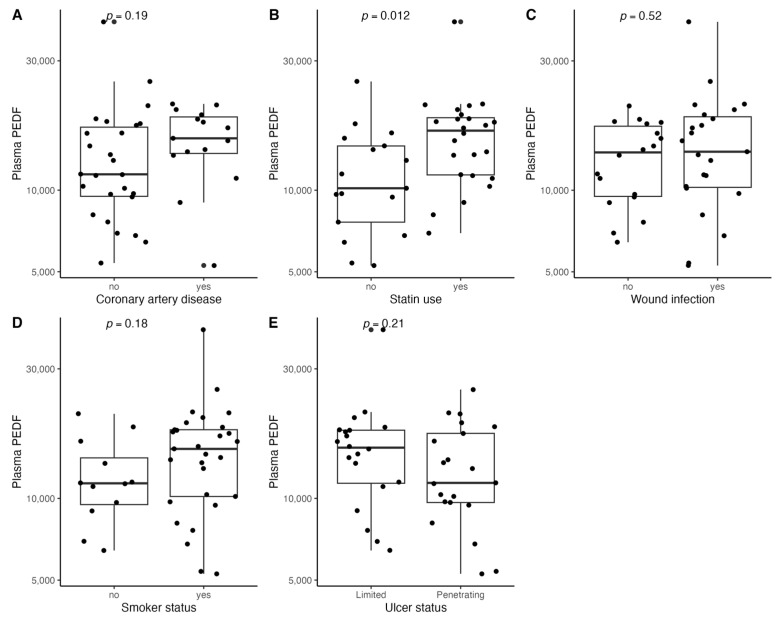
Boxplots with jittered data illustrating the relationships between plasma PEDF and (**A**) coronary artery disease, (**B**) statin use, (**C**) wound infection, (**D**) smoking status (yes—active or past smoker), and (**E**) ulcer depth PEDF—pigment epithelium-derived factor.

**Figure 12 biomedicines-11-01559-f012:**
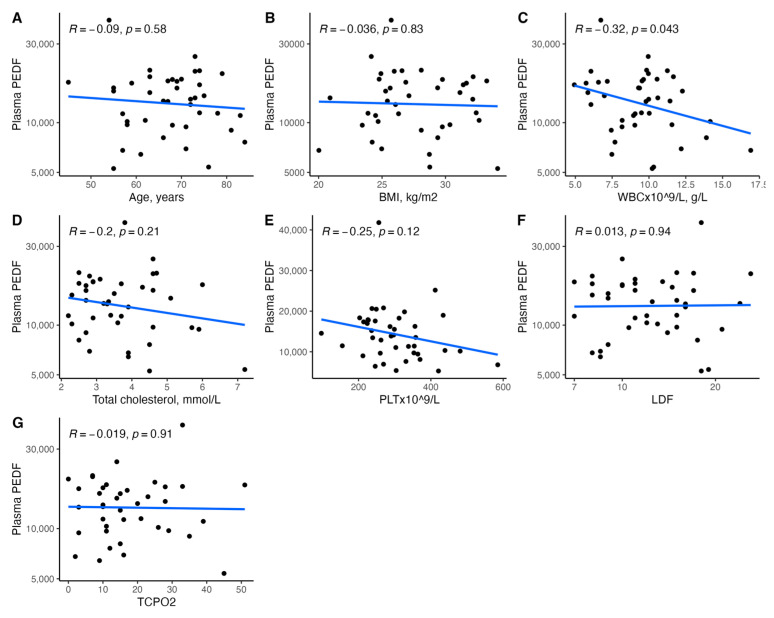
Scatterplots with a linear fit illustrating the relationships between plasma PEDF and (**A**) age, (**B**) BMI, (**C**) WBC, (**D**) total cholesterol, (**E**) PLT, (**F**) LDF, and (**G**) tcpO2. PEDF—pigment epithelium-derived factor, BMI—body mass index, WBC—white blood cells count, PLT—platelet count, LDF—laser Doppler flowmetry, and tcpO2—transcutaneous oximetry test.

**Figure 13 biomedicines-11-01559-f013:**
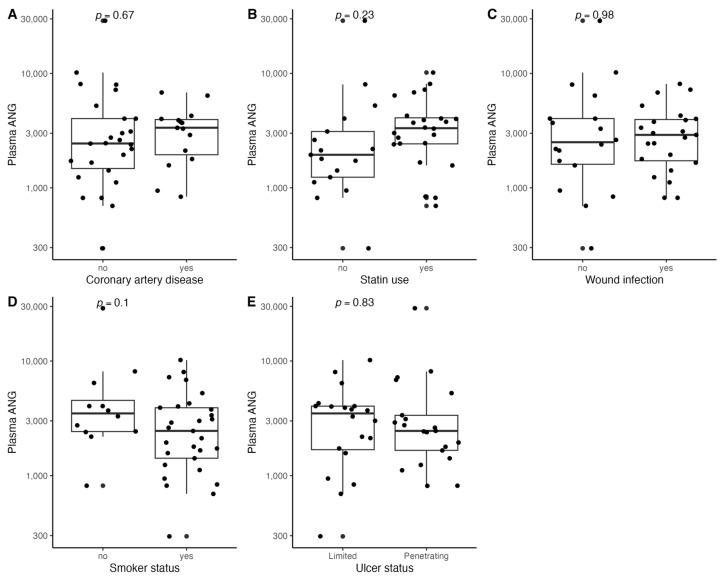
Boxplots with jittered data illustrating the relationships between plasma ANG1 and (**A**) coronary artery disease, (**B**) statin use, (**C**) wound infection, (**D**) smoking status (yes—active or past smoker), and (**E**) ulcer depth ANG1—angiopoietin-1.

**Figure 14 biomedicines-11-01559-f014:**
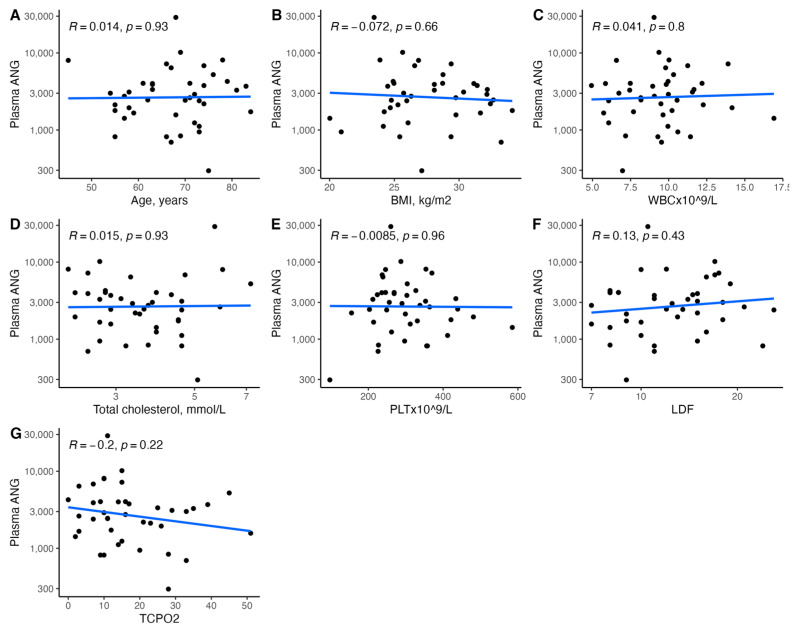
Scatterplots with a linear fit illustrating the relationships between plasma ANG1 and (**A**) age, (**B**) BMI, (**C**) WBC, (**D**) total cholesterol, (**E**) PLT, (**F**) LDF, and (**G**) tcpO2. ANG-1—angiopoietin-1, BMI—body mass index, WBC—white blood cells count, PLT—platelet count, LDF—laser Doppler flowmetry, and tcpO2—transcutaneous oximetry test.

**Figure 15 biomedicines-11-01559-f015:**
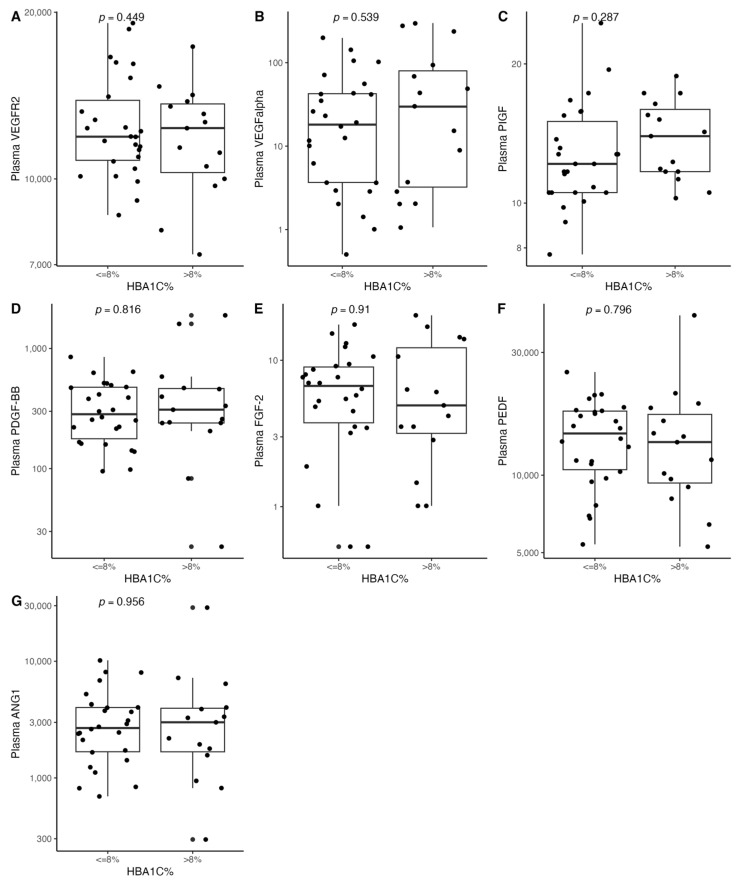
Box and jitter plots illustrating the relationships between (**A**) VEGFR2, (**B**) VEGF-A, (**C**) PIGF, (**D**) PDGF-BB, (**E**) FGF-2, (**F**) PEDF, and (**G**) ANG-1 and glycemic control (HBA1c cut-off at 8%). VEGF-A—vascular endothelial growth factor A, VEGF-R2—vascular endothelial growth factor receptor 2, FGF-2—fibroblast growth factor 2, PlGF—placental growth factor, PDGF-BB—platelet-derived growth factor-BB, PEDF—pigment epithelium-derived factor, and ANG-1—angiopoietin-1.

**Figure 16 biomedicines-11-01559-f016:**
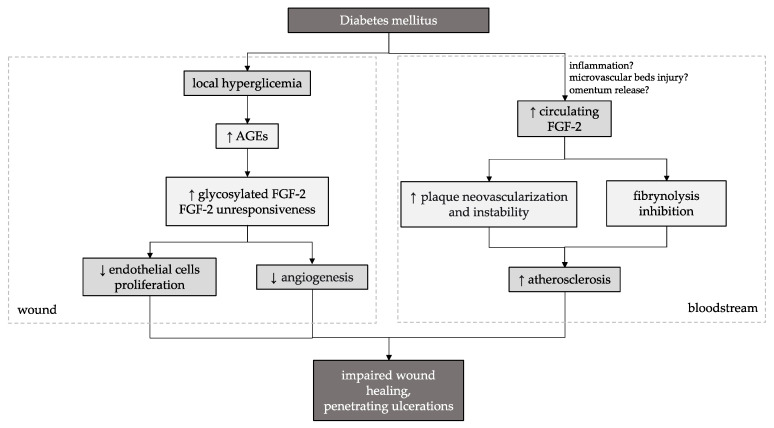
A diagram illustrating the potential double pathway of FGF-2 action on impaired wound healing in the course of DFS. ↑ increased, ↓ decreased.

**Table 1 biomedicines-11-01559-t001:** Concentrations of angiogenic and inflammatory markers in the plasma of patients with DFS compared to ulceration depth. Data are presented as mean (SD) unless otherwise indicated.

Variable	Limited ^1^ (n = 20)	Penetrating ^2^ (n = 21)	Total	*p*-Value
WBC, 10^9^/L	8.62 (1.99)	10.17 (2.71)	9.41 (2.48)	0.05
lnPLT	5.50 (0.28)	5.81 (0.28)	5.66 (0.32)	<0.01
lnCRP	1.35 (1.08)	2.73 (1.46)	2.06 (1.45)	<0.01
lnVEGF-A	2.55 (1.58)	3.02 (1.88)	2.79 (1.73)	0.40
VEGF-A, median (IQR), pg/mL	17.19 (2.92, 45.05)	22.91 (6.23, 101.44)	19.07 (3.67, 55.64)	
lnVEGF-R2	9.41 (0.21)	9.39 (0.23)	9.40 (0.22)	0.75
VEGF-R2, median (IQR), pg/mL	12,133.70 (10,700.38, 13,589.43)	11,927.98 (10,432.89, 14,082.89)	11,927.98 (10,539.92, 13,801.05)	
lnPIGF	2.63 (0.22)	2.50 (0.23)	2.56 (0.23)	0.09
PIGF, median (IQR), pg/mL	13.35 (11.66, 15.78)	11.70 (10.54, 13.74)	12.28 (10.83, 15.51)	
lnPDGF-BB	5.61 (0.81)	5.76 (0.77)	5.69 (0.79)	0.54
PDGF-BB, median (IQR), pg/mL	288.89 (194.36, 473.09)	299.64 (220.94, 469.20)	299.64 (215.23, 471.75)	
FGF-2, pg/mL	5.23 (4.17)	8.86 (5.29)	7.09 (5.06)	0.02
FGF-2, median (IQR), pg/mL	4.33 (2.61, 6.65)	7.66 (5.27, 13.06)	6.07 (3.52, 9.42)	
lnPEDF	9.57 (0.43)	9.39 (0.44)	9.48 (0.44)	0.21
PEDF, median (IQR), ng/mL	15,374.60 (11,369.19, 17,827.56)	11,403.50 (9656.45, 17,358.40)	13,870.45 (9720.83, 17,802.82)	
lnANG-1	7.85 (0.87)	7.91 (0.85)	7.88 (0.85)	0.83
ANG-1, median (IQR), pg/mL	3490.94 (1685.21, 4032.60)	2453.53 (1661.96, 3354.17)	2744.49 (1661.96, 4032.60)	

^1^ Confined to the skin and subcutaneous tissue. ^2^ Reaching the muscle, tendon, or more profound. ln—natural logarithm, PLT—platelet count, WBC—white blood cell count, CRP—C-reactive protein, VEGF-A—vascular endothelial growth factor A, VEGF-R2—vascular endothelial growth factor receptor 2, FGF-2—fibroblast growth factor 2, PlGF—placental growth factor, PDGF-BB—platelet-derived growth factor-BB, PEDF—pigment epithelium-derived factor, and Ang-1—angiopoietin-1. IQR—Interquartile Range.

**Table 2 biomedicines-11-01559-t002:** Hemodynamic parameters of lower limb arteries and microvascular status compared to ulceration depth. Data are presented as mean (SD) unless otherwise indicated.

Variable	Limited ^1^ (n = 20)	Penetrating ^2^ (n = 21)	Total	*p*-Value
ABI	1.10 (0.81)	0.73 (0.57)	0.91 (0.71)	0.11
ABI, median (IQR)	0.73 (0.59, 1.48)	0.67 (0.44, 0.81)	0.72 (0.46, 1.00)	
TBI	0.16 (0.14)	0.18 (0.09)	0.17 (0.12)	0.69
TBI, median (IQR)	0.15 (0.07, 0.19)	0.16 (0.11, 0.21)	0.15 (0.11, 0.21)	
tcpO2, mmHg	20.70 (13.06)	13.95 (10.36)	17.32 (12.13)	0.08
tcpO2, median (IQR), mmHg	18.50 (11.50, 29.25)	11.00 (8.50, 15.25)	14.50 (9.75, 25.25)	
LDF, PU	11.57 (3.56)	14.48 (5.07)	13.06 (4.59)	0.04
LDF, median (IQR), PU	10.50 (8.50, 15.00)	13.00 (11.00, 17.50)	12.00 (9.00, 16.00)	

^1^ Confined to the skin and subcutaneous tissue. ^2^ Reaching the muscle, tendon, or more profound. ABI—ankle–brachial index, TBI—toe–brachial index, tcpO2—transcutaneous oximetry test, and LDF—laser Doppler flowmetry. IQR—Interquartile Range, PU—perfusion units.

## Data Availability

Data sharing is not applicable to this article.

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
