# Peer review of "Circulating Angiogenic Factors and Ischemic Diabetic Foot Syndrome Advancement—A Pilot Study"

_biomedicines, 2023, doi:10.3390/biomedicines11061559_

Round 1

Reviewer 1 Report

In global terms, the manuscript presented by Schönborn et al. It is an idea that can present something new within your area. However, very notable aspects must be improved:

-In the first place, the manuscript is written in a confusing language and without a scientific order. In general terms, the authors must completely rewrite the manuscript.

-The summary of the manuscript must be rewritten, in its current state it seems like a popular article and not a scientific manuscript.

-The authors should improve the introduction to the state of the art, focusing on the mechanisms of the disease itself and justifying the need for the study.

-Authors should review and update references.

-The methodology section should be explained better.

-The authors must justify the sample size statistically.

-Authors must calculate the statistical potential of the sample size.

-Authors must adequately specify the inclusion criteria. The authors should improve the characterization of the patients in Table 1.

-In global terms, the authors should better describe the results in all points.

-The authors should improve the quality of the figures and unify the format.

-Authors must show data as interquartile range.

-Authors should divide the four figures into more and should be more self-explanatory.

-Authors must include a control.

-Aging must be a key point. Authors must stratify patients based on this very important criterion.

-The authors should improve the discussion with more clinical aspects in order to justify the translation of the study.

-Authors should improve the use of English grammar with professionals.

-Authors must include a graphic summary.

Reviewer 2 Report

Overall, the paper titled "Circulating angiogenic factors and ischemic diabetic foot syndrome advancement – a pilot study" presents interesting findings on the potential role of circulating angiogenic factors in the progression of ischemic diabetic foot syndrome (DFS). However, there are some areas that could be improved.

Firstly, the authors should clearly state their proposed future trends for research in this area. While they suggest that further studies with a larger patient population are needed to confirm their hypothesis regarding FGF-2 as a predictor for DFS advancement, they do not elaborate on other potential directions for future research in this field.

Secondly, the authors should provide a more thorough comparison of their results with published data on this topic. While they briefly mention a few studies that have investigated the relationship between angiogenic factors and DFS, they do not provide a comprehensive overview of the existing literature in this area. This would help to place their findings in a broader context and highlight any novel or unexpected results.

Thirdly, the authors could consider submitting a graphical scheme or diagram to accompany their abstract, illustrating the pathways or mechanisms they hypothesize to be involved in the relationship between angiogenic factors and DFS. This would provide readers with a visual representation of the study's main findings and could help to clarify complex concepts.

Round 2

Reviewer 1 Report

The authors have substantially improved the manuscript. However, the quality of the figures remains poor and difficult to follow. Please, the authors should improve the presentation of the figures. A very dramatic point is the use of English grammar, the authors must extensively improve the manuscript in this regard with native experts.

English very difficult to understand/incomprehensible

Reviewer 2 Report

The authors have performed a positive improvement.

Author Response

Dear Reviewer,

Thank you very much for your response. We are glad that our corrections met all expectations.